

# Ambient temperature and cardiovascular mortality: a systematic review and meta-analysis

Mohammad Taghi Moghadamnia[1], Ali Ardalan[1,2,*], Alireza Mesdaghinia[3,*], Abbas Keshtkar[4], Kazem Naddafi[3] and Mir Saeed Yekaninejad[5]

[1] Department of Disaster Public Health, School of Public Health, Tehran University of Medical Sciences, Tehran, Iran

[2] Harvard Humanitarian Initiative, Harvard University, Cambridge, United States of America

[3] Department of Environmental Health Engineering, School of Public Health, Tehran University of Medical Sciences, Tehran, Iran

[4] Department of Health Sciences Education Development, School of Public Health, Tehran University of Medical Sciences, Tehran, Iran

[5] Department of Epidemiology and Biostatistics, School of Public Health, Tehran University of Medical Sciences, Tehran, Iran

[*] These authors contributed equally to this work.

Corresponding author
Ali Ardalan, aardalan@tums.ac.ir

## ABSTRACT

**Introduction.** Our study aims at identifying and quantifying the relationship between the cold and heat exposure and the risk of cardiovascular mortality through a systematic review and meta-analysis.

**Material and Methods.** A systematic review and meta-analysis were conducted based on the Preferred Reporting Items for Systematic Reviews and Meta-Analyses (PRISMA) guideline. Peer-reviewed studies about the temperature and cardiovascular mortality were retrieved in the MEDLINE, Web of Science, and Scopus databases from January 2000 up to the end of 2015. The pooled effect sizes of short-term effect were calculated for the heat exposure and cold exposure separately. Also, we assessed the dose–response relationship of temperature-cardiovascular mortality by a change in units of latitudes, longitude, lag days and annual mean temperature by meta-regression.

**Result.** After screening the titles, abstracts and full texts, a total of 26 articles were included in the meta-analysis. The risk of cardiovascular mortality increased by 5% (RR, 1.055; 95% CI [1.050–1.060]) for the cold exposure and 1.3% (RR, 1.013; 95% CI [1.011–1.015]) for the heat exposure. The short-term effects of cold and heat exposure on the risk of cardiovascular mortality in males were 3.8% (RR, 1.038; 95% CI [1.034–1.043]) and 1.1% ( RR, 1.011; 95% CI [1.009–1.013]) respectively. Moreover, the effects of cold and heat exposure on risk of cardiovascular mortality in females were 4.1% (RR, 1.041; 95% CI [1.037–1.045]) and 1.4% (RR, 1.014; 95% CI [1.011–1.017]) respectively. In the elderly, it was at an 8.1% increase and a 6% increase in the heat and cold exposure, respectively. The greatest risk of cardiovascular mortality in cold temperature was in the 14 lag days (RR, 1.09; 95% CI [1.07–1.010]) and in hot temperatures in the seven lag days (RR, 1.14; 95% CI [1.09–1.17]). The significant dose–response relationship of latitude and longitude in cold exposure with cardiovascular mortality was found. The results showed that the risk of cardiovascular mortality increased with each degree increased significantly in latitude and longitude in cold exposure (0.2%, 95% CI [0.006–0.035]) and (0.07%, 95% CI [0.0003–0.014]) respectively. The risk of cardiovascular

abstract
mortality increased with each degree increase in latitude in heat exposure (0.07%, 95% CI [0.0008–0.124]).

**Conclusion**. Our findings indicate that the increase and decrease in ambient temperature had a relationship with the cardiovascular mortality. To prevent the temperature-related mortality, persons with cardiovascular disease and the elderly should be targeted. The review has been registered with PROSPERO (registration number CRD42016037673).

## INTRODUCTION

The relationship between climate change and health is considered to be a major concern in the health care system (*Cheng & Su, 2010*). Exposure to hot temperature is associated with physiological changes which include: increased plasma viscosity and cholesterol levels in serum (*Lin et al., 2013b*). Some studies have demonstrated that ambient temperature has effects on overall mortality (*Huang et al., 2012*; *Lin et al., 2013b*; *Rocklöv, Kristie & Bertil, 2010*). The relationship between ambient temperature and mortality has been described as a J, V, or U-pattern (*Huang, Wang & Yu, 2014*; *Lin et al., 2013b*; *Ryti, Guo & Jaakkola, 2015*).

The relationship between temperature, morbidity, and mortality is mediated by direct and indirect ecological processes (*Habib, El Zein & Ghanawi, 2010*). The majority of deaths due to heat waves affect individuals with pre-existing cardiovascular disease (*Pfeiffer, 2011*). Some studies have further shown that increases in cardiovascular diseases (CVD) is associated with both cold and hot temperatures (*Bhaskaran et al., 2009*).

Acute coronary syndrome (ACS) usually occurs in the late stages of coronary heart disease (CHD) and is one of the leading causes of death in the world (*Li et al., 2011*). The significant inverse relation between temperatures and acute coronary syndrome (ACS) incidence suggests that coldness may play an important role in the incidence of ACS. Some studies have reported greater mortality in cold weather than heat-related mortality (*Gomez-Acebo, Llorca & Dierssen, 2013*; *Medina-Ramón & Schwartz, 2007*). In addition, the heat wave effects appear for a short time whereas the effects of cold spells may persist for up to two months (*Goggins et al., 2013*; *Xie et al., 2012*). The harmful effects of cold and heat are strongly apparent in CVDs (*Zhou et al., 2014*). New research indicates that global warming, increasing ambient temperatures, and heat-related effects are of particular importance in the public health perspective (*Turner et al., 2012*).

The effects of temperature based on individual, social, economic and geographic factors are different. The poor and the elderly with underlying medical conditions, and those living in big cities are considered particularly vulnerable (*Ng et al., 2014*). The relationship between mortality and temperature may be dependent on the latitude. Thus, it is necessary that some studies be carried out in areas of different latitudes (*Ahmadnezhad et al., 2013*; *Shonkoff, 2012*).

Moghadamnia et al. (2017), *PeerJ*, DOI 10.7717/peerj.3574

2/27

Therefore, our study aims at identifying and quantifying the relationship between cold and heat exposure and risk of cardiovascular mortality through a systematic review and meta-analysis.

## MATERIALS AND METHOD

A systematic review and meta-analysis were conducted based on the Preferred Reporting Items for Systematic Reviews and Meta-Analyses (PRISMA) guidelines.

### Search strategy and sources

Peer-reviewed studies about temperature and cardiovascular mortality were retrieved in the databases which included: MEDLINE (via PubMed); Web of Science; and Scopus, from January 2000 up to the end of 2015. Reference lists of the selected studies were scanned in order to identify any further studies. To access important "gray literature we examined the following websites: the World Health Organization; the Intergovernmental Panel on Climate Change; the National Institutes of Health (USA) and the Department for Environment, Food, and Rural Affairs (UK).

### Search keywords and terms

We conducted a systematic search in two phases. In the primary investigation, the following keywords were used: ("weather" or "climate") and "cardiovascular diseases". All sub-terms were also included but we limited the search only to human studies published in English.

Then secondary search similar to primary was followed although we used more specific terms for CVDs such as "myocardial infarct*", "coronary event", "heart attack", "Q wave infarct*", "non-Q wave infarct*", "Acute coronary syndrome", "QWMI", "NQWMI", "STEMI", "NSTEMI", "coronary infarct*", "heart infarct*", "myocardial thrombosis", "coronary thrombosis", "congestive heart failure" and "heart failure". The specific terms for ("weather" or "climate") were used if included "temperature", " ambient temperature", "cold temperature", "hot temperature", where "*" indicates any word ending.

### Selection of articles

The following eligibility criteria were used in this study:

1. Studies applying time-series regression research design and case-crossover study design.
2. The nonlinear model statistical analysis (DLNM or GAM) in the time series studies. These methods show the effect of exposure event to be distributed over a specific period of time, using several parameters in explaining the contributions at different lags, thus providing a comprehensive picture of the time-course of the exposure outcome relationship cannot use the linear estimators and a non-linear one is appropriate. This method was first developed by Gasparrini and colleagues in 2011 (*Gasparrini, 2011*; *Gasparrini, Armstrong & Kenward, 2010*; *Lin et al., 2013b*).
3. Studies observing mortality outcomes in all types of CVDs.
4. Those reporting results which included the effect estimate associated with one unit increase or decrease in temperature. An outcome measure was reported as either: relative risk, odds ratio, regression coefficient or percent change.

5. Also, the studies which used the percent change in mortality associated with one unit increase or decrease in the temperature by turning odds ratio with this formula; $(OR - 1) \times 100 =$ percent increase.

On the other hand, the exclusion criteria were:

1. The studies providing only linear curves of temperature–outcome relationship
2. The studies using indoor temperature as the exposure variable
3. The qualitative studies without precise statistical relationship
4. Studies related to periods of extreme temperature such as the cold spells and heat waves.

## Data extraction

The results of titles and abstracts of all relevant studies were merged into the Endnote software, and the duplicates were removed. Abstracts and titles were screened by two independent reviewers and non-relevant articles were deleted. Full-text articles which met the inclusion criteria for the systematic review were downloaded. We were able to obtain full-text papers of all the potentially eligible studies and there was no need to contact their corresponding authors. The results of reviews were then compared and in the case of any discrepancies, they were resolved through consensuses. We tested publication bias using Egger's Method.

## Study quality assessment

In order to separate low-quality studies from others, quality assessment was necessary. Assessing the quality of public health studies and their risk of bias may be difficult. This is partly due to the wide variety of study designs used. Assessment of the quality or risk of bias for all of the reviewed studies was conducted by two independent reviewers using the Critical Appraisal Skills Program (*CASP, 2014*). This was adapted from Critical Appraisal Skills Program, Public Health Resource Unit, and Institute of Health Science, Oxford (*Oxman et al., 1994*). In this checklist, the quality score ranged from 1 to 10. A cut-off point of 6 was used for any study to be included in this systematic review and meta-analysis. The quality of studies was investigated based on: the study design/time-span; the population of study; sample size; statistical methods; main temperature exposure variable; confounder variables and lag time.

## Statistical analysis

Statistical analysis was performed in two steps. At first, the pooled effect size was computed for temperature exposure to the cold and hot weather using the random-effects meta-analysis. In the next step, dose–response effects of temperature on cardiovascular mortality were modeled using the random effects meta-regression.

In all of the reviewed studies, the hot and cold effect were calculated separately in six studies using the percent change in mortality associated with one unit increase or decrease of the temperature. The odds ratio (OR) were calculated using this formula; $(OR - 1) \times 100 =$ percent increase mortality. Then we converted the ORs to relative risk (RRs) using the following equation: $RR = OR/[(1 - P0) + (P0 \times OR)]$, where P0 = the incidence of non-exposed group. When the value of P0 is extremely small, we assume $RR = OR$.

In one study (*Goggins et al., 2013*), the effect of a 10 °C decrease of temperature on cardiovascular mortality was reported. Here, in order to calculate the risk of CVD mortality per 1 °C decrease of temperature, first, the percent of death due to cardiovascular mortality was changed to the relative risk then this effect size was divided by 10.

In some studies comparison between several percentiles used for the heat and cold temperature effects on cardiovascular mortality which had a great influence on mortality in our analysis. All the studies provided different lag patterns estimating the exposure-outcome and the delayed effect of temperature change from a single day to 35 days. In this case, we chose the largest estimate effects.

In cases where the results of the temperature effects on mortality in one city observed in two studies were similar, only one study was analyzed. If the outcomes of cardiovascular mortality in the two studies were different, then both were included. The pooled effect sizes were calculated separately for the cold and heat temperature. Also, we analyzed pooled effect size based on subgroup for males, females and vulnerable age groups in cold and hot temperature separately.

In order to evaluate the city-specific estimates, we calculated the $I^2$ criteria. It was used to investigate heterogeneity between studies, whereby the increasing values (from 0% to 100%) could explain the increasing heterogeneity (*Turner et al., 2012*). We used the Bayesian hierarchical approach to pool the city-specific effect estimates.

At the end, to show the reason of high heterogeneity in the estimated effect sizes, meta-regression models were estimated. The dependent variables were the effect sizes (risk of cardiovascular mortality) of cold or heat exposure in each study. The explanatory variables were the mean annual temperature of the location of studies, the number of lag days in each study, latitude, and longitude of the location of studies. The meta-regression models were estimated using the study variance estimator. All the analyses were conducted using the Stata 12.0 (Stata Corporation, College Station, Texas, USA).

## RESULTS

As shown in Fig. 1, 681 articles were identified in the initial search. After screening the title and abstract, 626 articles were excluded because of not absolutely full-filling the inclusion criteria.

Of the remaining 55 articles, 10 were excluded due to their estimated effect based on extreme event weather such as cold spells and heat waves (*Barnett et al., 2012*; *Chen et al., 2015*; *Lee et al., 2015*; *Ma et al., 2015*; *Sheridan & Lin, 2014*; *Son et al., 2012*; *Tian et al., 2013*; *Xie et al., 2012*; *Zeng et al., 2014*; *Zhou et al., 2014*). Five studies were excluded based on the special meteorological indicators [(diurnal temperature range, (DTR)] (*Ding et al., 2015*; *Yang et al., 2013*), [universal thermal climate index (UTCI)] (*Burkart et al., 2014*), [Temperature Changes between Neighboring Days] (*Lin et al., 2013a*) and [Apparent Temperature] (*Sun et al., 2012*). In the eight studies with case-crossover design using special meteorological indicators (e.g., Apparent Temperature and Tapp$_{max}$) were excluded (*Basu & Ostro, 2008*; *Bell et al., 2008*; *Gronlund et al., 2015*; *Madrigano et al., 2013*; *Medina-Ramón & Schwartz, 2007*; *Stafoggia et al., 2006*; *Wichmann et al., 2011*; *Wilson et al., 2013*). Two studies provided the estimates of temperature effect on all- cause mortality, not

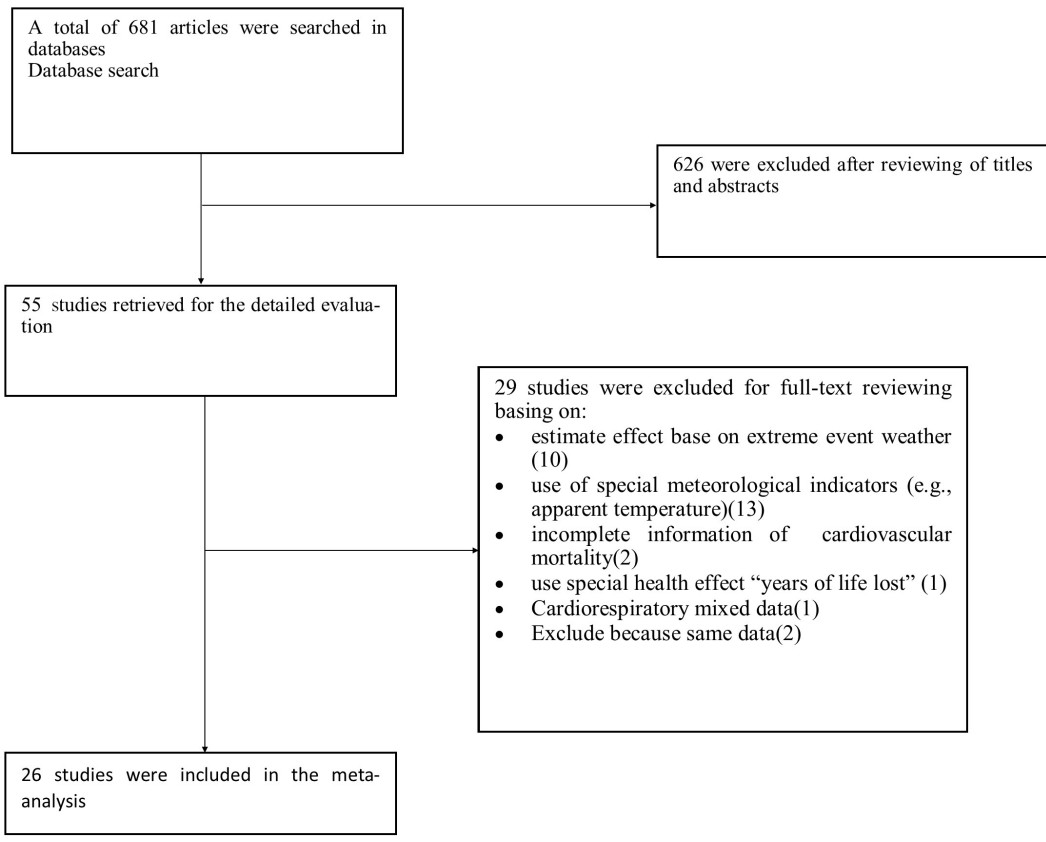

**Figure 1** Procedure for literature search.

special cardiovascular mortality (*Son et al., 2012*; *Vutcovici, Goldberg & Valois, 2014*), and one study used the particular effect measure entitled "years of life lost" (*Huang et al., 2012*). The data of one study on cardiovascular and respiratory mortality were mixed and not reported separately (*Goldberg et al., 2011*). Two studies written by the same authors, at the same time, geographical area and with similar information were removed (*Breitner et al., 2014b*; *Yu et al., 2011a*). The remaining 26 studies were included in the meta-analysis.

Among the included studies, 22 studies assessed effects of both heat and cold on cardiovascular mortality (*Bai, Woodward & Liu, 2014*; *Breitner et al., 2014a*; *Breitner et al., 2014b*; *Chen et al., 2014*; *Guo et al., 2012*; *Guo, Punnasiri & Tong, 2012*; *Huang, Wang & Yu, 2014*; *Huang et al., 2015*; *Kim et al., 2015*; *Lin et al., 2013b*; *Seposo, Dang & Honda, 2015*; *Tian et al., 2012*; *Wang et al., 2014*; *Wang et al., 2015*; *Wichmann et al., 2013*; *Yang et al., 2015a*; *Yang et al., 2012*; *Yang et al., 2015b*; *Yang et al., 2015c*; *Yi & Chan, 2015*; *Yu et al., 2011b*; *Yu et al., 2011c*). Three studies specifically examined the effect of heat on mortality (*Chan et al., 2012*; *Kim et al., 2016*; *Qiao et al., 2015*) and two studies examined the cold effects on mortality (*Goggins et al., 2013*; *Gomez-Acebo, Llorca & Dierssen, 2013*). The majority of studies utilized confounding variables such as $PM_{10}$, $O_3$, $NO_2$, $O_2$, humidity, atmospheric pressure, time trend and season leading to more reliable estimates of temperature effects on cardiovascular mortality. The day lag in considered studies ranged from 0 to 35

days. It was an important consideration that the effects of temperature on mortality extend beyond one day. The study period of included investigations was from two years (*Gomez-Acebo, Llorca & Dierssen, 2013*) till 31 years (*Kim et al., 2016*). A large number of studies used a specific percentile of temperature to test the presence of the heat and cold effect (*Bai, Woodward & Liu, 2014*; *Breitner et al., 2014a*; *Breitner et al., 2014b*; *Chen et al., 2014*; *Goggins et al., 2013*; *Gomez-Acebo, Llorca & Dierssen, 2013*; *Guo et al., 2012*; *Guo, Punnasiri & Tong, 2012*; *Kim et al., 2015*; *Lin et al., 2013b*; *Ma, Chen & Kan, 2014*; *Seposo, Dang & Honda, 2015*; *Tian et al., 2012*; *Wang et al., 2014*; *Wang et al., 2015*; *Yang et al., 2015a*; *Yang et al., 2012*; *Yang et al., 2015b*; *Yang et al., 2015c*; *Yi & Chan, 2015*). In the remaining studies, the specific heat or cold temperature was used as a starting point (*Chan et al., 2012*; *Huang, Wang & Yu, 2014*; *Huang et al., 2015*; *Kim et al., 2016*; *Qiao et al., 2015*; *Yu et al., 2011b*; *Yu et al., 2011c*). In most of the studies, the daily mean temperature was used as a main temperature exposure variable. In the study where researchers used mean temperatures and diurnal temperature range (DTR) as the thermal index, the mean temperature findings were included in the meta-analysis (*Kim et al., 2016*). In all of the included studies, 25 had time series design and one study case-crossover design. The final articles included in this analysis are listed in Table 1. In this table, comprehensive details on each study in terms of time, location, the type of heart disease and defined warm and cold temperature are presented.

The pooled effect sizes of the short-term effect of temperature exposure and cardiovascular mortality were separately reported for the cold and heat exposure.

The risk of cardiovascular mortality increased by 5% (RR, 1.055; 95% CI [1.050–1.060]) for the cold exposure, 1.3% (RR, 1.013; 95% CI [1.011–1.015]) for the heat exposure (Figs. 2 and 3). The short-term effects of cold and heat exposure on risk of cardiovascular mortality in males were 3.8% (RR, 1.038; 95% CI [1.034–1.043]) and 1.1% (RR, 1.011; 95% CI [1.009–1.013]) respectively (Figs. 4 and 5). Also effects of cold and heat exposure on risk of cardiovascular mortality in female were 4.1% (RR, 1.041; 95% CI [1.037–1.045]) and 1.4% (RR, 1.014; 95% CI [1.011–1.017]) respectively (Figs. 6 and 7). In the elderly, these figures raised to 8.1% and 6% in the heat exposure and cold exposure respectively (Figs. 8 and 9). The lag days have also an effect on the risk of cardiovascular mortality for the heat and cold exposure so that the greatest risk of cardiovascular mortality in a cold temperature was in the 14 lag days (RR, 1.09; 95% CI [1.07–1.010]) and in a hot temperature in the seven lag days (RR, 1.11; 95% CI [1.08–1.14]) (Table 2). Regarding the effect of low and high educational level on risk of CVD mortality, similar findings were revealed (Table 2).

The dose–response effects of the mean annual temperature, the number of lag days, latitude and longitude of the location of studies on the risk of cardiovascular mortality are shown in Table 3. The results showed that the temperature associated increase in risk of cardiovascular mortality increased with each degree increased significantly in latitude and longitude in cold exposure (0.2%, 95% CI [0.006–0.035] and (0.07%, 95% CI [0.0003–0.014]) respectively. The temperature associated increase in risk of cardiovascular mortality increased with each degree increase in latitude in heat exposure (0.07%, 95% CI [0.0008–0.124]). However, this dose–response relationship was not statistically significant in other explanatory variables. For both heat and cold exposure, $I^2$ values were mostly on

Moghadamnia et al. (2017), *PeerJ*, DOI 10.7717/peerj.3574

**Table 1 Characteristic of included studies.**

| NO | Authors and years of publication | Events No /Data source | Location and time period | Main temperature exposure variable (s) | Variables Controlled | Lags (Days) | Study design | Effect estimate of temperature/threshold (definition of hot & cold effect) | Outcome measurement |
|----|----|----|----|----|----|----|----|----|----|
| 1 | Lin et al. (2013b) | 1.253.75 mortality per day/Department of Health | Four regions of Taiwan 1994–2007 | Mean temperature | PM[a] 10, NO[b]$_x$, O[c] 3 | 0–20 | Time-Series | 15 °C compared with 27 °C for cold effect vs. 31 °C compared with 27 °C for hot effect | ICD[d]-9 Codes Ischemic heart disease and CVD |
| 2 | Tian et al. (2012) | 26,460/Death Classification System, Beijing Public Security Bureau | Beijing, China 2000–2011 | Daily mean temperature | Day of the week | 0–15 | Time-Series | 99th (30.5 °C) compared to 90th (27.0 °C) for hot effect vs 1st (−7.6 °C) compared to 10th (−2.2 °C) for cold effect | ICD-10: (I20–I25). CHD mortality |
| 3 | Yu et al. (2011b) | 22,805/Office of Economic and Statistical Research of the Queensland Treasury | Queensland, Australia 1996–2004 | Mean temperature | Time trend, PM$_{10}$, RH[e],NO$_2$, O$_3$ | 0–20 | Time-Series | 1 °C above 24 °C for hot effect 1 °C below 24 °C for cold effect | ICD-9: (390–459) ICD-10( I00-I79) CVD |
| 4 | Qiao et al. (2015) | 22,561/Office of Economic and Statistical Research of the Queensland Treasury | Brisbane, Australia 1996–2004 | Daily mean temperature | Time trend, seasonality | 0–20 | Time-Series | 1 °C mean temperature increase above the threshold (28 °C) | ICD-9 :(390–459) ICD-10(I00-I99) CVD |
| 5 | Goggins et al. (2013) | Hong Kong 91/d and Taiwan 33/d/Hong Kong Census and Statistics Bureau, Taiwan's Department of Health | Hong Kong 19999–2009 Taiwan 1999–2008 | Mean temperature | RH, PM$_{10}$, NO$_2$, SO2[f], O3 wind speed, solar radiation, Time trend, seasonality, Day of Week | 0–35 | Time-Series | 10 °C drop in temperature in cold seson for Cold effect | ICD-9 :(390–459) ICD-10(I00-I99) |
| 6 | Jun et al., 2015 (Yang et al., 2015c) | 1,936,116/Death Register and Report of Chinese CDC[g] | China 2007–2013 2007–2013 | Mean temperature | Not mentioned | 0–21 | Time-Series | 99th compared with MMT for hot effect vs1th compared with MMT for cold effect | ICD-10(I00-I99) |
| 7 | Kim et al. (2016) | Ranged from 3.3–50.5 mean daily/Chinese CDC Ministry of Health and Welfare of Japan, and the National Death Registry of Taiwan | 30 different cities of East Asia, 1979–2010 | Mean Temperature diurnal Temperature rang | PM$_{10}$, NO$_2$, and SO2 | 0–21 | Time-Series | 1 °C increase above mean temperature for hot effect | ICD-10(I00-I99) |

Moghadamnia et al. (2017), *PeerJ*, DOI 10.7717/peerj.3574

**Table 1** (*continued*)

| NO | Authors and years of publication | Events No /Data source | Location and time period | Main temperature exposure variable (s) | Variables Controlled | Lags (Days) | Study design | Effect estimate of temperature/threshold (definition of hot & cold effect) | Outcome measurement |
|---|---|---|---|---|---|---|---|---|---|
| 8 | *Yang et al. (2012)* | Guangzhou Bureau of Health | Guangzhou, China 2003–2007 | Maximum, Minimum and Mean temperature | $PM_{10}$, $NO_2$, and SO2 | 0–25 | Time-Series | 99th to the 90th for hot effect | ICD-10(I00-I99) Cardiovascular mortality |
| 9 | *Guo, Punnasiri & Tong (2012)* | 11,746/Bureau of Policy and Strategy, Ministry of Public Health, Thailand | Thailand 1999–2008 | Maximum, Minimum and Mean Temperature | PM10, O3, RH Influenza | 0–21 | Time-Series | 99th compared to 75th for hot effect vs 1st compared to 25th for cold effect | ICD-10(I00-I99) Cardiovascular mortality |
| 10 | *Yi & Chan (2015)* | 98,091/Hong Kong Census and Statistics Department | Hong Kong 2002–2011 | Maximum, Minimum and Mean Temperature | $PM_{10}$, $NO_2$, and SO2 | 0–21 | Time-Series | 99th compared to 75th for hot effect vs 1st compared to 25th for cold effect | ICD-10(I00-I99) Cardiovascular mortality |
| 11 | *Seposo, Dang & Honda (2015)* | 14.7 mortality per day Philippine Statistics Authority-National Statistics Office (PSA-NSO) | Philippine 2006–2010 | Daily average temperature | Seasonal effect | No | Time-Series | 1st respective to MMT for cold effect vs. 99th respective to MMT for hot effect | ICD codes (I00–I99) Cardiovascular-related mortality |
| 12 | *Chen et al. (2014)* | 126,925/Death Register System from Chinese CDC | China 2009–2011 | Maximum, Minimum and Mean Temperature | $PM_{10}$, $NO_2$, and SO2 | 0–14 | Time-Series | 99th compared to 75th for hot effect vs 1st compared to 25th for cold effect | ICD -10: (I00–I25) Coronary artery Disease |
| 13 | *Guo et al. (2012)* | 16,559/Chinese CDC | China 2004–2008 | Maximum, Minimum and Mean Temperature | $PM_{10}$, and NO2 | 0–20 | Time-Series | 99th compared to 90th for hot effect vs 1st compared to 10th for cold effect | ICD -10: (I00–I25) Coronary artery Disease |
| 14 | (*Kim et al., 2015*) | 8.5 mortality per day Statistics Korea, National Institute of Environmental Research | Korea 1995–2011 | Daily Mean Temperature | RH, holidays, Day of Week, Time trend, $PM_{10}$, NO2 | 0–21 | Time-Series | 99th compared to 90th for hot effect vs 10th compared to 25th for cold effect | ICD-10 (I20-I59) ICD-9(410–429) |
| 15 | (*Huang et al., 2015*) | 552,866/Chinese CDC | China 2006–2011 | Mean Temperature | Seasonal trend, Day of Week, RH, Duration of sunshine, Precipitation, Atmospheric Pressure | 0–21 | Time-Series | 1 °C increase from 25 °C for hot effect, 1 °C decrease from 25 °C for cold effect | ICD codes (I00–I99) |

Moghadamnia et al. (2017), *PeerJ*, DOI 10.7717/peerj.3574

**Table 1** (*continued*)

| NO | Authors and years of publication | Events No /Data source | Location and time period | Main temperature exposure variable (s) | Variables Controlled | Lags (Days) | Study design | Effect estimate of temperature/threshold (definition of hot & cold effect) | Outcome measurement |
|---|---|---|---|---|---|---|---|---|---|
| 16 | *Yang et al. (2015a)* | 57,806/Central urban district of Shanghai | Shanghai, China 1981-2012 | Daily mean Temperature | Seasonality, Day of Week RH, Holidays, population size | 0–28 | Time-Series | 99th compared to 90th for hot effect vs 1st compared to 10th for cold effect | ICD-9: (390–459) ICD codes (I00–I99) |
| 17 | *Breitner et al. (2014a)*; *Breitner et al. (2014b)* | Not mentioned/Bavarian State Office for Statistics and Data Processing | Bavaria, Germany 1990–2007 | Mean Temperature | Ozone, PM10 Influenza epidemic, time trend, Day of week, RH, barometric pressure | 0–14 | Time-Series | 99th compared to 90th for hot effect vs 1st compared to 10th for cold effect | ICD-9: (390–459)ICD codes (I00–I99) |
| 18 | *Wang et al. (2014)* | 18,530/Suzhou Center for Disease Control and Prevention | Suzhou, China 2005–2008 | Mean Temperature | PM$_{10}$, NO$_2$, and SO2 | 0–28 | Time-Series | 99th compared to MMT(26 C) for hot effect vs 1st compared to MMT(26 C) for cold effect | ICD codes (I00–I99) |
| 19 | (*Ma, Chen & Kan, 2014*) | Not mentioned/Municipal Center for Disease Control and Prevention (CDC) | 17 large cities of China 1996–2008 | Mean Temperature | PM$_{10}$, NO$_2$, and SO2 | 0–28 | Time-Series | 99th compared to 75th for hot effectvs 1st compared to 25th for cold effect | ICD codes (I00–I99) |
| 20 | *Yang et al. (2015b)* | 23.8 mortality per day Guangzhou Bureau of Health | Guangzhou, China 2003–2007 | Mean Temperature | PM10, NO2, and SO2, Seasonality, RH, Atmospheric Pressure | 0–30 | Time-Series | 99th compared to 75th for hot effect vs 1st compared to 25th for cold effect | ICD codes (I00–I99) |
| 21 | *Huang, Wang & Yu (2014)* | 19,418/Chinese CDC | Changsha, China 2008–2011 | Daily Mean, Maximum and Minimum temperature | Long-term, Seasonality, barometric pressure, RH | 0–30 | Time-Series | 1 °C decrease from 10 °C for cold effect vs 1 °C increase from 29 °C for hot effect | ICD codes (I00–I79) |
| 22 | *Yu et al. (2011a)* | 22,805/Office of Economic and Statistical Research of the Statistical Research of the Queensland Treasury | Brisbane, Australia 1996–2004 | Maximum, Minimum Temperature | PM10, NO2, O3 | 0–31 | Time-Series | 1 °C increase from 24 °C for cold effect vs 1 °C decrease from 24 °C for cold effect | ICD-9: (390–499)ICD codes (I00–I99) |
| 23 | *Chan et al. (2012)* | 129,688/Hong Kong Census And Statistics Department | Hong Kong 1998–2006 | Average daily mean temperature | PM10, NO2, SO2, O3 Day of the week and holiday | 0–14 | Time-Series | 1 °C increase from 28.2 °C for hot effect | ICD9, 390–459 ICD 10 I00–I99 |
| 24 | *Wang et al. (2015)* | Not mentioned/Chinese CDC | Chinese 2007–2009 | Mean Temperature | Seasonality, Time trend, PM10, NO2, SO2, RH, Wind speed | 0–27 | Time-Series | 99th compared to 90th for hot effect vs 1st compared to 10th for cold effect | ICD 10 I00–I99 |

Moghadamnia et al. (2017), *PeerJ*, DOI 10.7717/peerj.3574

Peer J

**Table 1** (*continued*)

| NO | Authors and years of publication | Events No /Data source | Location and time period | Main temperature exposure variable (s) | Variables Controlled | Lags (Days) | Study design | Effect estimate of temperature/threshold (definition of hot & cold effect) | Outcome measurement |
|----|----|----|----|----|----|----|----|----|----|
| 25 | *Bai, Woodward & Liu (2014)* | 5,610/Tibetan CDC | China 2008–2012 | Mean Temperature | – | 0–14 | Time-Series | 99th compared to 75th for hot effect vs 1st compared to 10th for cold effect | ICD 10 I00–I99 |
| 26 | *Gomez-Acebo, Llorca & Dierssen (2013)* | 1,252/Spanish National Institute for Statistical | Spain 2004–2005 | Minimum temperatures | Age, sex, underlying disease | 0–6 | Case-crossover | 5th (−13.8 C) compared with over the 5th (1.8 C) for cold effect | ICD-10 I00–I99 |

**Notes.**

[a]PM: Particle Matter.
[b]NOx: Nitrous Oxide.
[c]$O_3$: Ozone.
[d]ICD: International Classification of Diseases.
[e]RH: Relative Humidity.
[f]$SO_2$: Sulfur Dioxide.
[g]CDC: Center of control Diseases.

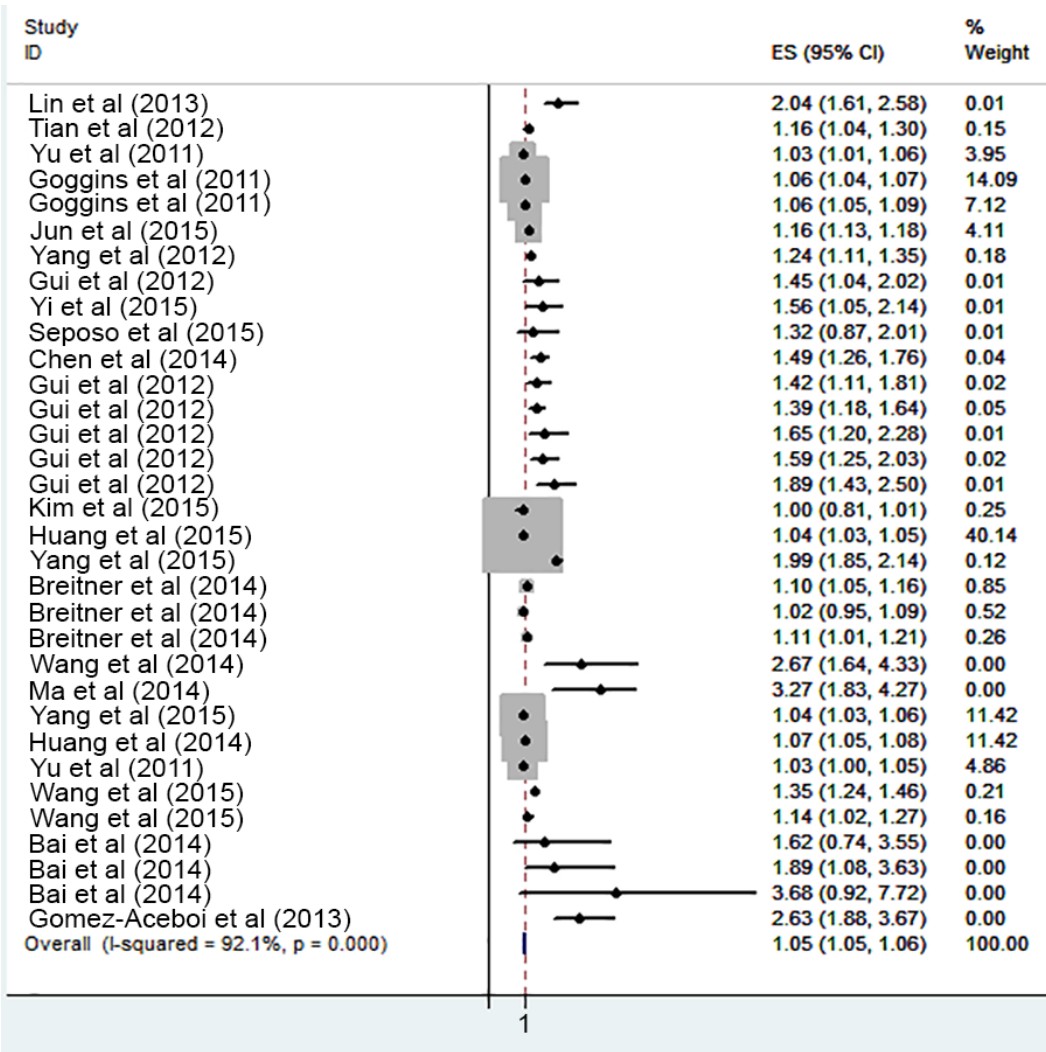

**Figure 2   Meta-analysis of ambient temperature on risk of cardiovascular mortality in cold exposure.**

the order of 76%–92%, indicating large heterogeneity among studies, and emphasizes the use of random-effects models.

## DISCUSSION

To the best of our knowledge, this study is the first meta-analysis to survey the association between temperature and cardiovascular mortality. Also, findings of this study showed that both cold and hot temperatures increased the risk of cardiovascular mortality, although a cold temperature had stronger effects on CVD mortality. This result is consistent with other investigations that revealed cold weather as responsible for the most part of the temperature-related CVD mortality (*Goggins et al., 2013*; *Xie et al., 2012*; *Yang et al., 2015c*). In contrast to our findings, *Grjibovski et al. (2012)* in a study in Astana, Kazakhstan-the second coldest capital in the world between 2000–2001 and 2006–2010, didn't find any significant associations between ambient temperatures and mortality of hypertensive

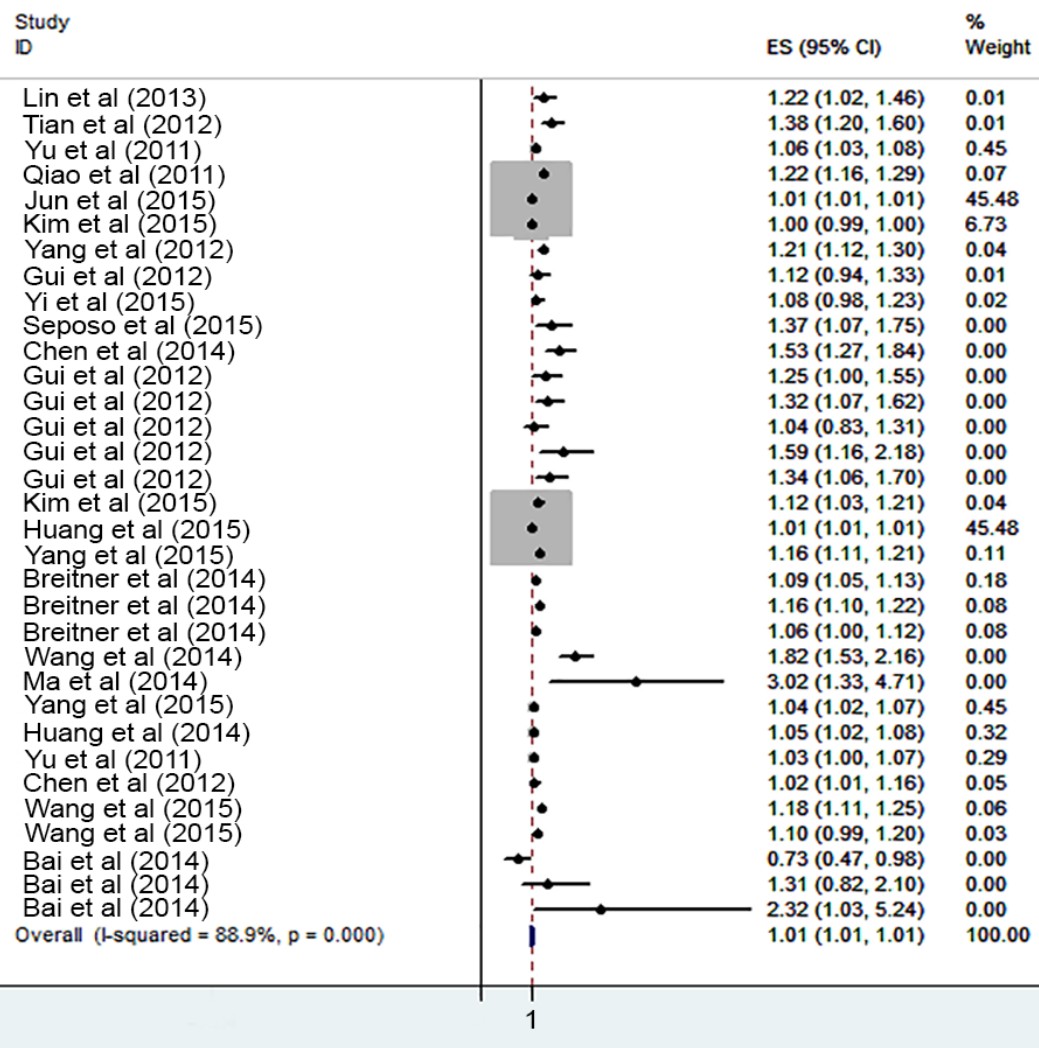

**Figure 3  Meta-analysis of ambient temperature on risk of cardiovascular mortality in heat exposure.**

diseases (ICD-10 codes: I10–I15), ischemic heart disease (I20–I25) and CVDs (I60–I69). The inconsistency between the findings may be due to differences in geographical location of studies and weather conditions.

The cold seasons causes physiological changes including increases in blood sugar, levels of cholesterol, fibrinogen concentration and platelet aggregation. Fibrinogen plays an important role in the formation of clots in the coronary artery, the start of acute myocardial infarction and life-threatening arrhythmias (*Čulić, 2007*; *Mittleman & Mostofsky, 2011*). Cold temperature causes peripheral vasoconstriction and increase of cardiac afterload in patients with preexisting CVDs as a result of exacerbating symptoms. Furthermore, cold weather may induce complications such as respiratory infections with an indirect effect on the cardiovascular performance.

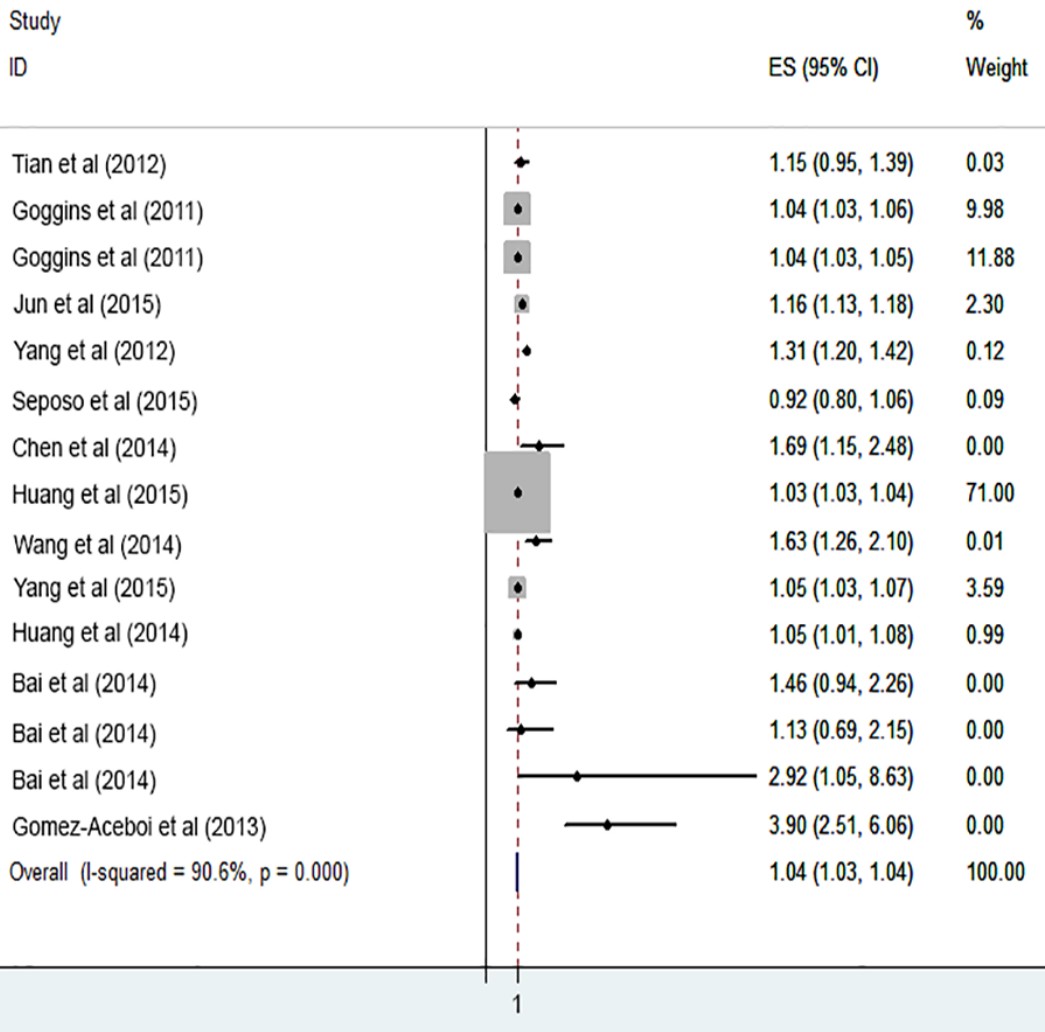

| Study ID | | ES (95% CI) | % Weight |
|---|---|---|---|
| Tian et al (2012) | | 1.15 (0.95, 1.39) | 0.03 |
| Goggins et al (2011) | | 1.04 (1.03, 1.06) | 9.98 |
| Goggins et al (2011) | | 1.04 (1.03, 1.05) | 11.88 |
| Jun et al (2015) | | 1.16 (1.13, 1.18) | 2.30 |
| Yang et al (2012) | | 1.31 (1.20, 1.42) | 0.12 |
| Seposo et al (2015) | | 0.92 (0.80, 1.06) | 0.09 |
| Chen et al (2014) | | 1.69 (1.15, 2.48) | 0.00 |
| Huang et al (2015) | | 1.03 (1.03, 1.04) | 71.00 |
| Wang et al (2014) | | 1.63 (1.26, 2.10) | 0.01 |
| Yang et al (2015) | | 1.05 (1.03, 1.07) | 3.59 |
| Huang et al (2014) | | 1.05 (1.01, 1.08) | 0.99 |
| Bai et al (2014) | | 1.46 (0.94, 2.26) | 0.00 |
| Bai et al (2014) | | 1.13 (0.69, 2.15) | 0.00 |
| Bai et al (2014) | | 2.92 (1.05, 8.63) | 0.00 |
| Gomez-Aceboi et al (2013) | | 3.90 (2.51, 6.06) | 0.00 |
| Overall  (I-squared = 90.6%, p = 0.000) | | 1.04 (1.03, 1.04) | 100.00 |

**Figure 4  Meta-analysis of cold exposure and risk of cardiovascular mortality in males.**

Our study also indicated that hot exposure increase risk of cardiovascular mortality. In this regard, *Qiao et al. (2015)* reported a sensible effect of high temperatures on cardiovascular deaths in summer.

Exposure to high temperature could increase the viscosity of plasma and cholesterol levels in serum (*Lin et al., 2013b*). Also, it has been shown that hot temperature leads to the increase in red blood cell and platelet counts (*Bhaskaran et al., 2009*). However, further studies are required to identify the accurate mechanisms of heat effect on the CVD mortality.

Our finding showed that both hot and cold temperature had a strong effect on cardiovascular mortality in elderly. In agreement with our finding, Yu et al. reported that 1 °C increase in temperature was associated with 2–5% increase in mortality rate whereas 1 °C decrease in temperature causes 1–2% increase in mortality (*Yu et al., 2012*). The ability to regulate body temperature decreases with age and an elevated sweating

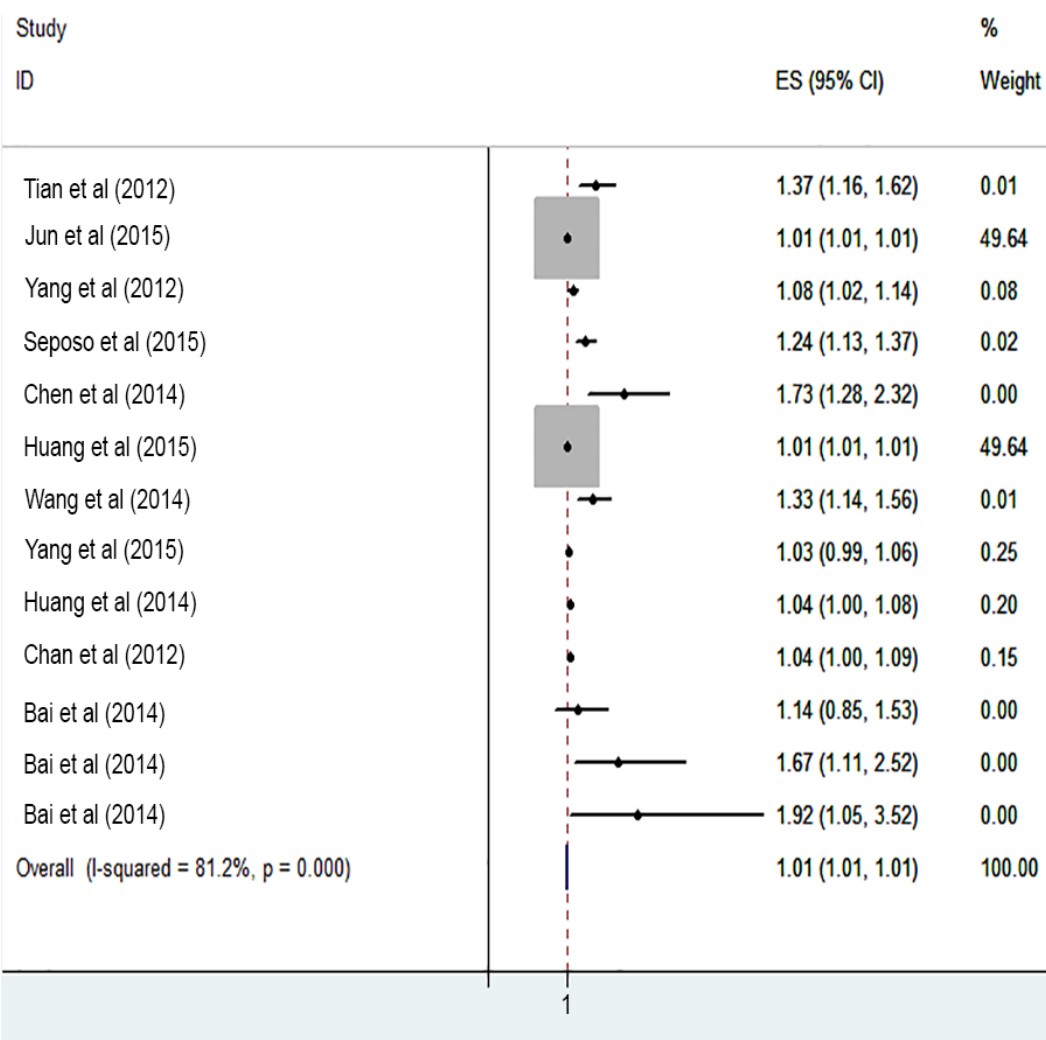

**Figure 5** Meta-analysis of heat exposure and risk of cardiovascular mortality in males.

threshold. In hot temperatures, older people may lose body fluids and become dehydrated. They, thus, eventually experience the cardiovascular complications. Moreover, the process of atherosclerosis escalates the conditions for developing CVDs such as ACS.

Results of this study showed that the effect of cold and hot temperature on cardiovascular mortality regarding the gender was approximately the same. In this regard, *Kan et al. (2007)* indicated that the effect of the cold and heat temperature was not significantly different between the two genders. However, the findings of *Yang et al. (2012)* and *Lim et al. (2015)* showed that men were less sensitive to effects of DTR on cardiovascular mortality and morbidity than women. Conversely, Chen and colleagues (*2014*) reported extreme temperatures were significantly associated with cardiovascular mortality among both males and females, but the associations were stronger for males.

We noticed that high and low temperature had the greatest effect in lag 7 and 14 respectively. According to the previous studies, the temperature variability had delayed

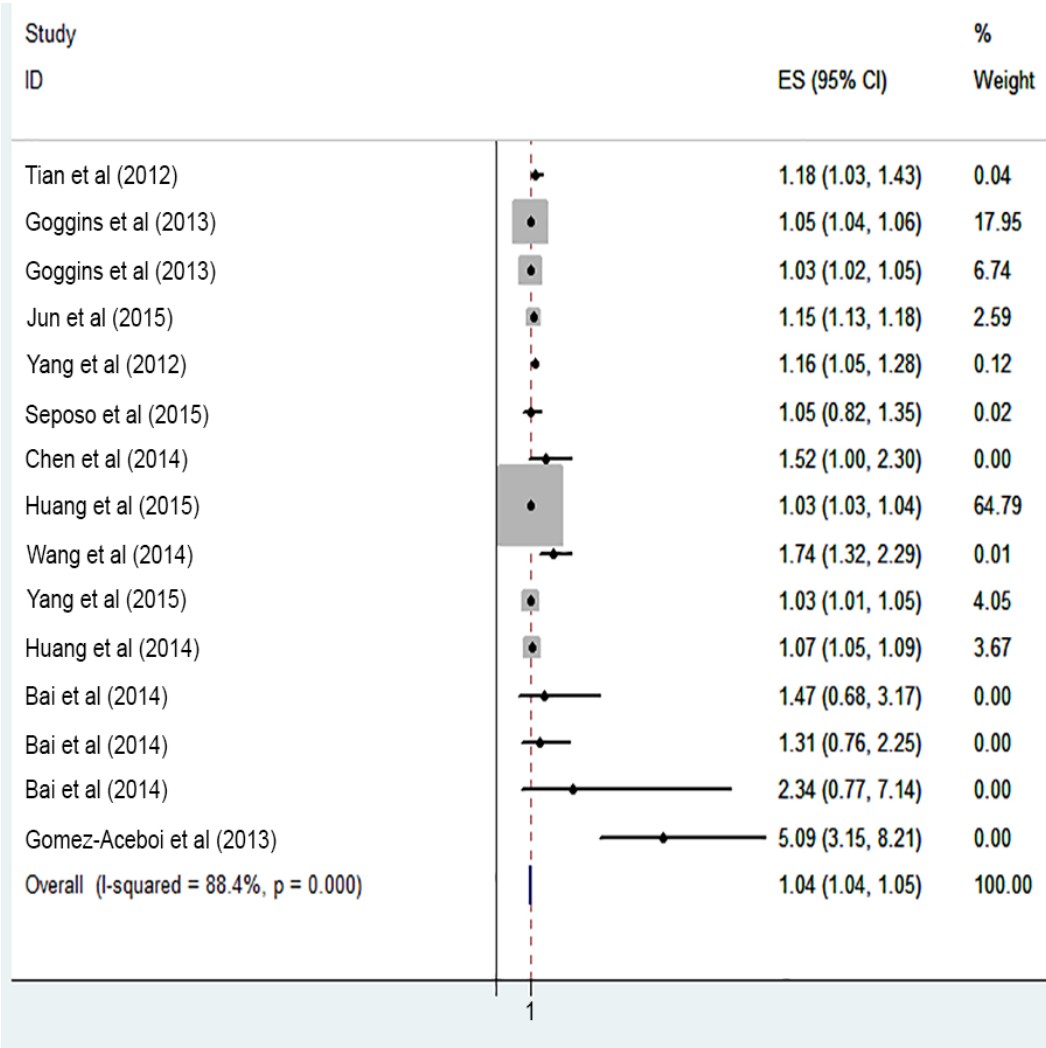

**Figure 6  Meta-analysis of cold exposure and risk of cardiovascular mortality in females.**

effect on health. The study conducted by *Huang, Wang & Yu (2014)* showed that effect of the cold temperatures had a long lag period of 10–25 days while the hot temperatures had a short lag period of only 1–3 days. Several recent studies reported that the cold days have a longer lag effect on the cardiovascular morbidity and mortality compared to hot days, (*Lin et al., 2013b*; *Yu et al., 2011b*; *Yu et al., 2011c*). However, to date, the underlying environmental and physiological mechanisms for various lag effects of heat and cold exposure remain unclear thus more studies are required.

Many factors may influence the relationship between ambient temperature and cardiovascular mortality either directly or indirectly as confounders. Our results did not show significant differences in mortality in people with different educational levels in hot and cold weather. In one study, results showed that the level of education and socioeconomic status could also be affecting the relationship between temperature and mortality (*Son et al., 2012*). The discrepancies between the findings of this study and others
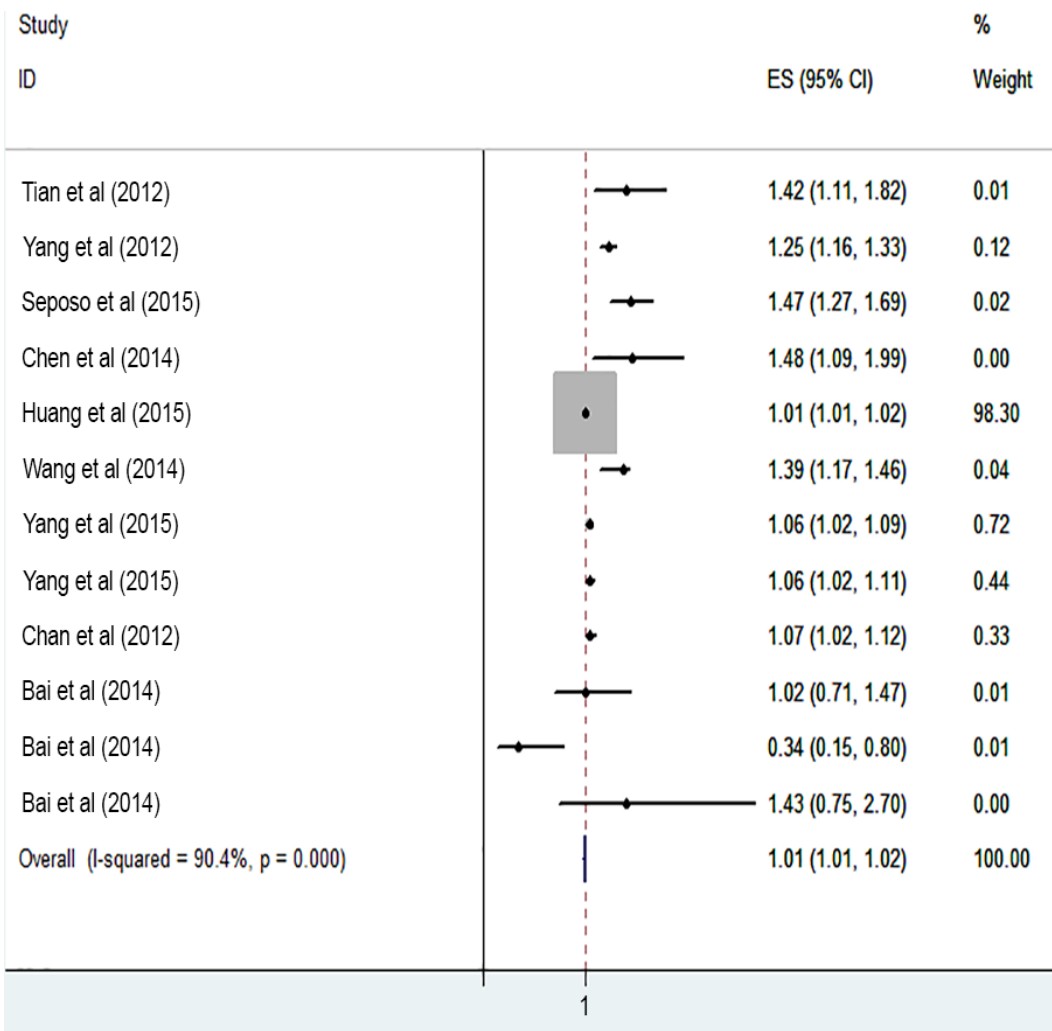

**Figure 7  Meta-analysis of heat exposure and risk of cardiovascular mortality in females.**

are perhaps due to the paucity of studies in this regard. The people living in low-income countries with poor access to suitable heating or cooling systems much experience the adverse effect of temperature. Also, those with a low level of education have been reported to be more vulnerable to exposure–response relationship.

For dose–response relationship between temperature and risk of CVD mortality, one-degree change for latitude significantly increased the risk of mortality. That means higher latitude countries show higher effects of cold temperature on the risk of cardiovascular mortality. Findings of many previous studies showed that extreme cold temperature is a major public health threat in high-latitude countries (*Kysely et al., 2009*; *Shaposhnikov et al., 2014*). This study revealed increase in longitude had higher effects of cold temperature in mortality. Differences of longitude may incorporate the different characteristics between cities and nations. Further study is required to assess how characteristics of cities and nations may modify the temperature effect. Also, a one-degree increase of latitude significantly

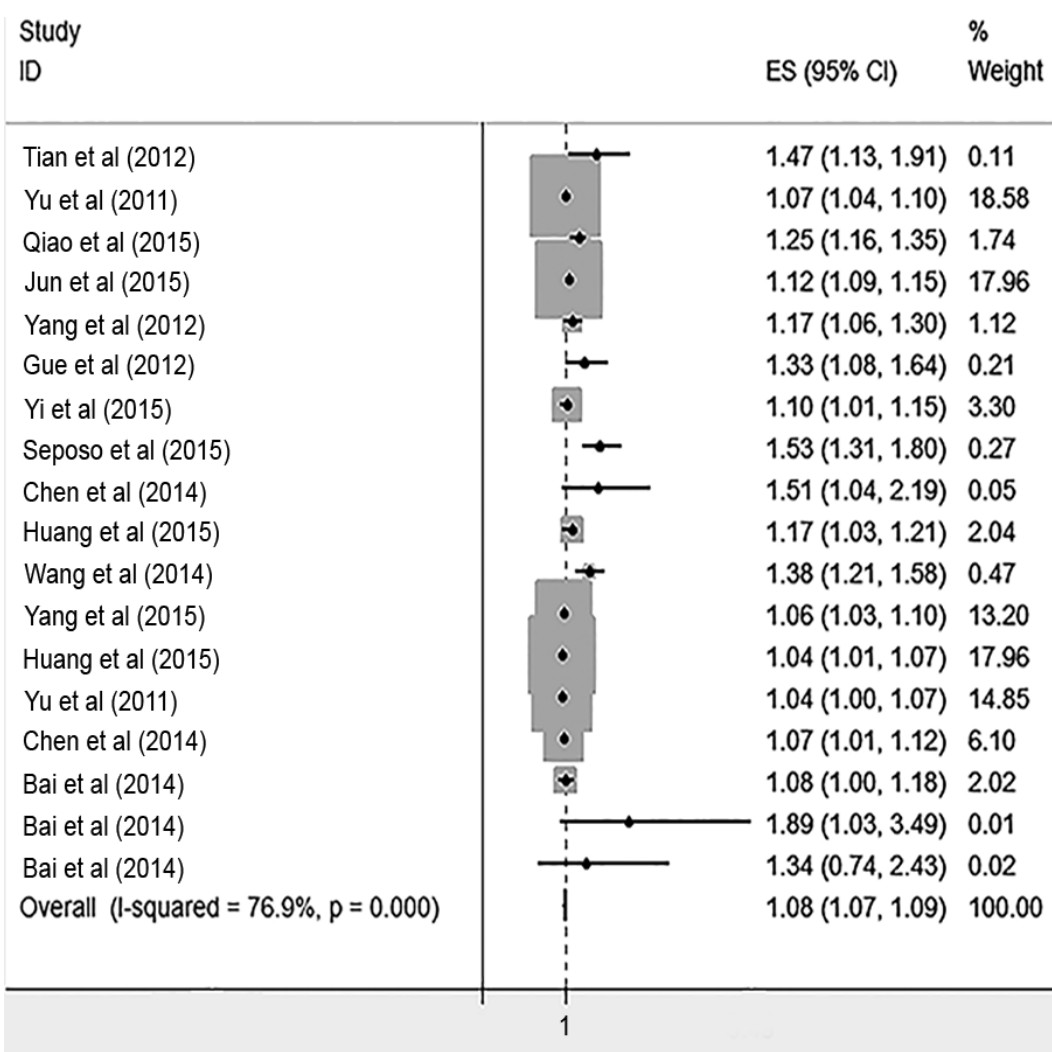

**Figure 8** Meta-analysis of heat exposure on risk of cardiovascular mortality in vulnerable age groups.

increased the risk of mortality in heat temperature. This finding is consistent with the previous reviews (*Kim et al., 2016*). The reasons for the latitude effect in colder weather were that the adaptive capacity of people who live in cold climate is lower because the population is less acclimatized to hot temperatures and live in houses that are improper for hot weather.

This study has several strengths. First, it is the first study to apply a systematic review and meta-analysis in assessing the available and valid literature related to the effects of ambient temperature on cardiovascular mortality. Second, we used the daily mean temperature which is the best indicator to show the exposure—outcome relationship of the cold and heat ambient temperature. In this case, previous studies only showed that the daily mean temperature was the best predictor of the death counts (*Anderson & Bell, 2009*). Finally, we included the studies with nonlinear models statistical analysis (DLNM). The major advantage of this is that it is flexible enough to simultaneously describe a non-linear

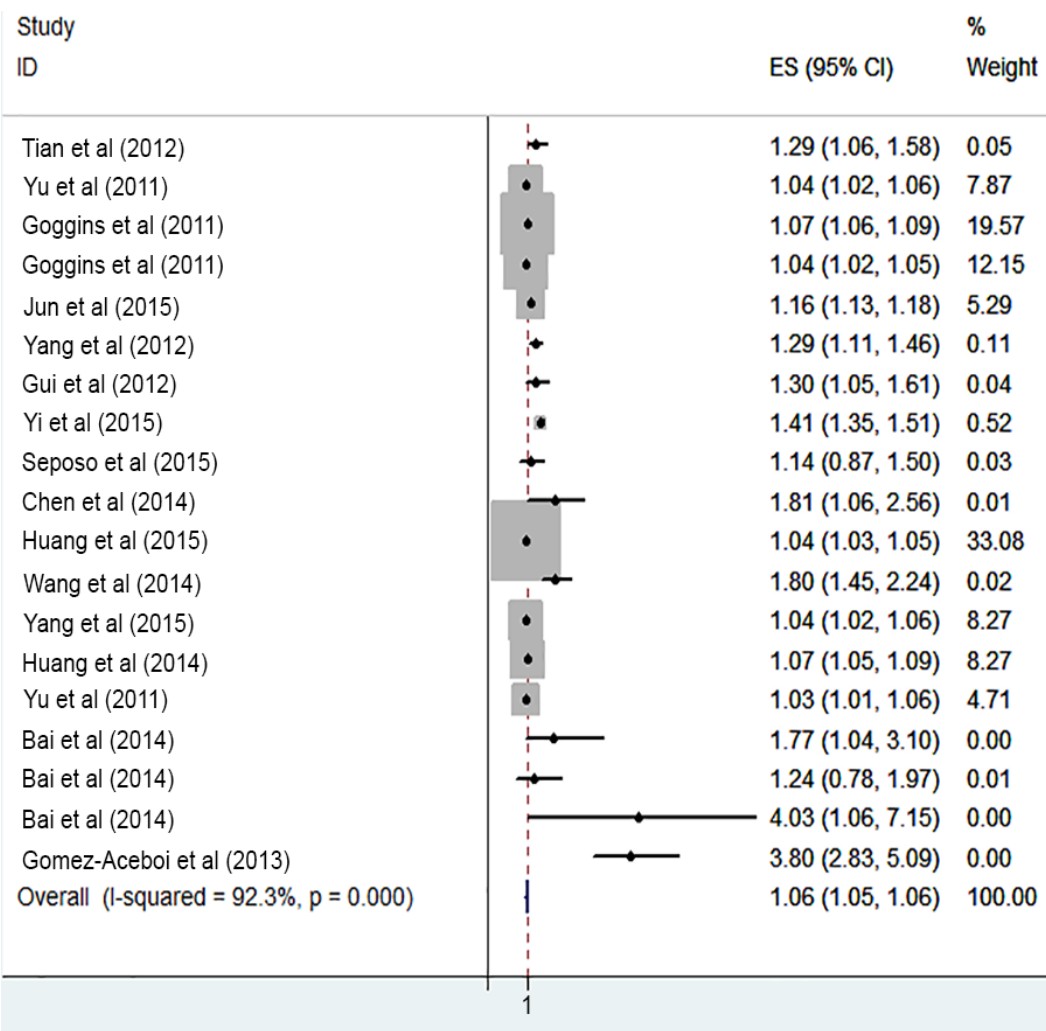

**Figure 9** Meta-analysis of cold exposure on risk of cardiovascular mortality in vulnerable age groups.

**Table 2** The cumulative relative risks (95% confidence interval) of cold and heat exposure on cardio-vascular mortality based on lag days and educational levels.

| Variabels | | Cold effect | Heat effect |
|---|---|---|---|
| Lag days | Lag 0 | 1.01 (1.00–1.02) | 1.04 (1.03–1.05) |
| | Lag 0–3 | 1.06 (1.05–1.08) | 1.10 (1.08–1.12) |
| | Lag 0–7 | 1.05 (1.03–1.07) | 1.14 (1.09–1.17) |
| | Lag 0–13 | 1.09 (1.07–1.10) | 1.11 (1.08–1.15) |
| | Lag 0–21 | 1.06 (1.05–1.07) | 1.12 (1.06–1.17) |
| | Lag 0–28 | 1.07 (1.05–1.08) | 1.13 (1.07–1.17) |
| Educational levels | Low Educational Level | 1.035 (1.031–1.04) | 1.03 (1.02–1.04) |
| | High Educational Level | 1.014 (1.011–1.02) | 1.011 (1.00–1.015) |

**Table 3** The dose–response relationship between the temperature associated increase in risk of cardiovascular mortality (%) and latitude, longitude, lag day and mean annual temperature.

| Explraitory variable | Cold exposure | | Heat exposure | |
|---|---|---|---|---|
| | Coefficient (95% Conf. Interval) | *P* value | Coefficient (95% Conf. Interval) | *P* value |
| One-degree change in latitude | 0.020 (0.0060–0.0356) | 0.009 | 0.007 (0.0008–0.0124) | 0.026 |
| One-degree change in longitude | 0.007 (0.0003–0.0146) | 0.042 | 0.0006 (−0.002–0.004) | 0.655 |
| One-day increase in lag | −0.0167 (−0.055–0.021) | 0.366 | 0.006 (−0.009–0.022) | 0.385 |
| One-degree increase in mean annual temperature | 0.026 (−0.019–0.072) | 0.240 | 0.008 (−0.0151–0.031) | 0.474 |
| Constant variable | 248.867 (17.507–480.227) | 0.037 | 175.327 (64.792–285.862) | 0.004 |

**Notes.**
* Significant at 95% CI.

exposure-response association and delayed effects or harvesting effect. This method was first developed by Gasparrini and colleagues in 2011 (*Gasparrini, 2011*; *Gasparrini, Armstrong & Kenward, 2010*; *Lin et al., 2013b*).

However, this study was not without some limitations. Although an accurate and complete search to select the eligible primary studies was employed, most of the studies reviewed were from the South East Asian countries such as China and few others from other parts of the world. Lack of attention to the use of air conditioning in summer, heating system in winter and socioeconomic and demographic factors in many primary studies that were considered for this review and analysis is also another likely limitation to the study.

## CONCLUSION

Our systematic review and meta-analysis showed that the ambient temperature is associated with increased cardiovascular mortality rates. According to our findings, the increase and decrease in the ambient temperature had a relationship with the cardiovascular mortality. People with underlying heart disease especially the elderly are more vulnerable to the cold and hot effects. To prevent the temperature-related mortality, people with cardiovascular disease and vulnerable groups, especially the elderly, should be targeted.

Based on the evidence rereviewed, most studies were conducted in the South East Asia so future studies are recommended in other parts with a focus on specific geographical and climate areas.

### Funding
The authors received no funding for this work.

### Competing Interests
The authors declare there are no competing interests.

## Author Contributions

- Mohammad Taghi Moghadamnia performed the experiments, contributed reagents/materials/analysis tools, wrote the paper, prepared figures and/or tables, reviewed drafts of the paper.
- Ali Ardalan conceived and designed the experiments, wrote the paper, reviewed drafts of the paper.
- Alireza Mesdaghinia conceived and designed the experiments.
- Abbas Keshtkar conceived and designed the experiments, analyzed the data, contributed reagents/materials/analysis tools.
- Kazem Naddafi and Mir Saeed Yekaninejad performed the experiments.

## Data Availability

The raw data has been supplied as a Supplementary File.

## Supplemental Information

Supplemental information for this article can be found online at http://dx.doi.org/10.7717/peerj.3574#supplemental-information.

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
