# Peer review of "Ambient temperature and cardiovascular mortality: a systematic review and meta-analysis"

_PeerJ, doi:10.7717/peerj.3574_

## Round 0.1 · original submission · Major Revisions

Please revise the paper according to the comments from reviewers or give reasonable rebuttal if you think the comments are not correct.

·

Basic reporting

This is one important study aiming at the CVD mortality effects of both cold and heat exposures. The study was clearly presented in a professional way, and the language is good. The structure of the manuscript is well organized.

The authors did a thorough literature review to retrieve relevant studies in the literature, and provided sufficient data on this topic.

Experimental design

The study aims, methods, results were well presented.
The authors clearly presented their research question, and methods to retrieve information from the literature.

Validity of the findings

The topic is not novel in this research field but with high public health significance.

Additional comments

This is an interesting study in the context of global climate change.

I have a few comments for the authors to consider to improve their study:
1. "basing on" should be "based on".
2. Not necessary to show the sex-specific results given that they are similar, would be fine to simply say that the effects were similar among males and females.
3. Could the authors consider giving a description on the definition of heat and cold in each of the studies, and whether the differences affect the pooled effect estimate.
4. also for the reference temperature each study used to examine the effects of cold and heat exposures.
5. did the author check whether the definition of young and old people is same in different studies?
6. I suggest that the author should make it clear that it is the short-term effect.
7. the author mentioned lag, I also suggest to differentiate the single day effect and cumulative effect of multi-days.
8. I would be more interesting to have one more analysis to examine the effect of location of the study area (latitude, longitude).
9. The mortality data quality would be one more factor to consider, such as the overall mortality rate (kind of indicator of completeness of the mortality data).

Reviewer 2 ·

Basic reporting

This paper performed an interesting systematic review and meta-analysis to investigate the relationship between ambient temperature and cardiovascular mortality. Although present study is important, of potential interest to readers, data are correctly analyzed and discussed, some points have to be considered.

Experimental design

1. Selection of articles: Explain the reasons for including the studies with nonlinear model statistical analysis and excluding the studies with linear model statistical analysis.
2. In the methodology section, the subgroup analysis method should be described in detail.

Validity of the findings

1. Line 208:This study investigated the relationship between ambient temperature and cardiovascular mortality. Why mention "morbidity"?
2. Discussion: The discussion is not thorough enough to explain the possible causes of the differences in the results of the different studies.
3. Conclusion: Providing suggestions for future research.

Additional comments

1. Random-effect: should be changed in 'random-effects' throughout the manuscript.

·

Basic reporting

While the English is understandable there are grammatical errors and having someone who is a native speaker go over the manuscript would improve the quality of the paper.

The Discussion section is rather disorganized. It would be good to organize paragraphs each with a distinct subject: for example: paragraphs on: highlights of main results, comparison to past studies, possible mechanisms behind temperature CVD associations.
The latter part of the Discussion on Strengths & limitations is fine.

Discussion Line 207-209: 'Although previous studies showed no significant association between ambient temperature and cardiovascular morbidity(Turner et al. 2012), our study tries to indicate a relationship between them'. It needs to be noted that the Turner et al study was a systematic review and also noted that they looked at morbidity rather than mortality. As your paper examines mortality it is not the same thing and therefore it's not trying to indicate a relationship. It is curious that heat is associated with CVD mortality but not morbidity. Perhaps the authors can try to speculate on possible reasons for this?

A study is mentioned in the Discussion, Grjibovski et al, but it is not included in the systematic review? Why was it not included?

Experimental design

Systematic review, meta-analysis and meta-regression. The presentation of the meta-analysis is confusing. The overall effects listed are presumably for 1C increases or decreases but for individual studies the effect size measurement varies. So for example our studies for Hong Kong and Taipei (number 5) used 10C decreases for the effect size. It would be clearer if the converted risk increases for 1C were used in the table.

Validity of the findings

The meta-regression results are also presented in a confusing way. The OR and SE are presented but it's unclear why OR is presented? Was logistic regression used? If so how were effect sizes dichotomized. If linear regression, which seems more appropriate, was used than coefficients should be reported. It is also better to report confidence intervals and P-values rather than SEs. Also why was region divided into SE Asia and others? Climate, particularly mean annual temperatures are more likely to be a predictor of effect sizes for temperature effects.

For my study the entry in table 1 is incorrect. The effect estimate is for a 10C drop in temperature, not for 'below 10th' as listed in the table. The authors need to check the these results carefully.

Additional comments

This is an interesting study and the authors have done a thorough job reviewing the literature. However they need to be do a major revision more carefully reporting the results and also re-do the meta-regression with some indicator of local climate as a predictor variable.

---

## Round 0.2 · Minor Revisions

The manuscript has been improved a lot. But there are still some very minor issues which should be revised before acceptance. Please make the edits noted by Reviewer 2 and take this opportunity to give the whole paper a final proof-read.

·

Basic reporting

All my comments have been addressed. I would suggest to accept this manuscript.

Experimental design

It used appropriate methods to do this study.

Validity of the findings

It is valid.

Additional comments

All my comments have been addressed. I would suggest to accept this manuscript.

·

Basic reporting

This is a re-review. I am satisfied with the revisions the authors have made and believe that the paper is ready for publication. I have just a couple of minor suggestions for further revision.

Experimental design

Fine.

Validity of the findings

Fine.

Additional comments

Minor revisions:

Abstract Results: ‘female’ -> ‘females’

Line 180, line 182, and Table 3: ‘risk’ -> ‘temperature associated increase in risk’.

---

## Round 0.3 · accepted · Accept

Congratulations on your publication at PeerJ.